# The Brahmavarta Initiative: A Roadmap for the First Self-Sustaining City-State on Mars

**Arvind Mukundan** [ID] **and Hsiang-Chen Wang** *[ID]

Department of Mechanical Engineering, Advanced Institute of Manufacturing with High Tech Innovations (AIM-HI) and Center for Innovative Research on Aging Society (CIRAS), National Chung Cheng Univesity, 168, University Rd., Min Hsiung, Chia Yi 62102, Taiwan
* Correspondence: hcwang@ccu.edu.tw

**Abstract:** The vast universe, from its unfathomable ends to our very own Milky Way galaxy, is comprised of numerous celestial bodies—disparate yet each having their uniqueness. Amongst these bodies exist only a handful that have an environment that can nurture and sustain life. The *Homo sapiens* species has inhabited the planet, which is positioned in a precise way—Earth. It is an irrefutable truth that the planet Earth has provided us with all necessities for survival—for the human race to flourish and prosper and make scientific and technological advancements. Humans have always had an innate ardor for exploration—and now, since they have explored every nook and corner of this planet, inhabiting it and utilizing its resources, the time has come to alleviate the burden we have placed upon Earth to be the sole life-sustaining planet. With limited resources in our grasp and an ever-proliferating population, it is the need of the hour that we take a leap and go beyond the planet for inhabitation—explore the other celestial objects in our galaxy. Then, however, there arises a confounding conundrum—where do we go? The answer is right next to our home—the Red Planet, Mars. Space scientists have confirmed that Mars has conditions to support life and is the closest candidate for human inhabitation. The planet has certain similarities to Earth and its proximity provides us with convenient contact. This paper will be dealing with the conceptual design for the first city-state on Mars. Aggregating assumptions, research, and estimations, this first settlement project shall propose the most optimal means to explore, inhabit and colonize our sister planet, Mars.

**Keywords:** Mars Colonization; self-sustaining city-state; life support system; terraforming

## 1. Introduction

### 1.1. About

This paper presents team Brahmavarta's[1] response to the challenge posed by The Mars Society to design a working model for the first self-sustaining human settlement on Mars. The city-state should be self-supporting to the maximum extent possible—i.e., relying on a minimum mass of imports from Earth. The goal is to have the city-state be able to produce all the food, clothing, shelter, power, common consumer products, vehicles, and machines for a population of 1,000,000 Brahmavartans[2], with only the minimum number of key components, such as advanced electronics, needed to be imported from Earth. This paper will explore both the technical and non-technical aspects involved in planning the first self-sustaining human settlement on Mars.

### 1.2. Phases of Brahmavarta Initiative

Despite their obvious differences, the red planet and blue planet do have some things in common. The rotation speed for one day on Mars is only about 44 min longer than a day on Earth [1]. Mars's axial tilt is also 25° compared to our planet's 23°; therefore, Mars will undergo similar seasonal and temperature variations [2]. Both the planets are made up of metallic cores and similar mineral compositions. They have similar surface

structures including mountains, canyons, and deserts [3]. But the differences between the two planets are much more significant. Martian atmosphere, unlike on Earth, is very thin, only measuring at about 1% of Earth's atmospheric pressure and being completely unbreathable for humans [4]. It is composed of 96% $CO_2$, 2% argon, and 2% nitrogen, with a trace amount of oxygen and water vapor [5]. Mars is drastically colder than Earth, averaging $-46\ ^\circ$C, with $-143\ ^\circ$C in the winter and $35\ ^\circ$C in summer on the equator [6]. Mars is also very dry and dusty and is buffeted by frequent sandstorms. Mars lacks any reasonably sized magnetosphere, measuring between 16 and 40 times weaker than Earth's, which leaves it more susceptible to harmful cosmic rays. Gravity clocks in at about 37% of that of Earth, which would be a small challenge to overcome, but nothing compared to the temperature and lack of atmosphere. Any successful Martian colony would have to contend with these big problems to fight to establish any kind of hospitable environment in what should be considered a world hostile to life. To counter all the obstacles, various parts of the literature were reviewed and assumptions were created around which The Brahmavarta Initiative has been developed. First, it is assumed that the technology for carrying 100 tons of payload to Mars with 1100 tons of propellant will be tested and ready to use by 2026 [7–9]. Liquid methane and liquid oxygen ($CH_4/O_2$) will be used as propellants, operating in mixture ratios between 3:1 and 3.5:1 (oxygen: methane) [10,11]. Secondly, the full support of Brahmavarta is assumed for the entirety of the mission. To better understand the challenges involved, the whole design for Brahmavarta is conceptually divided into the following phases: The Pre-Initiative phase, the Settlement phase, and the Self-Sustaining phase. Finally, the cost of shipping goods from Earth to Mars will be $500/kg, and the cost of shipping goods from Mars to Earth will be $200/kg.

## 2. Pre-Initiative Phase

### 2.1. Martian Administration on Earth (MADE)

By the year 2024, MADE will come into existence when governments and space agencies will join hands to sign the Mars treaty and will become the investors in this project. The number of Brahmavartans and the authorities on MADE will be based on the investments, with the highest being the greatest number of Brahmavartans. MADE is split into six different ministries, with the Martian Ambassador overlooking the operations with a flat hierarchy, as shown in Figure 1. The head of each ministry will act as a board of directors in decisions. In the case of even votes, the Martian ambassador will have another vote to decide. The board of directors will be responsible for managing the functionality of sub-departments. Goods arriving from Brahmavarta will be handled by the Ministry of External Affairs in exchange for resources with governments on Earth. The Ministry of Finance will handle the investments and the budgets for funding missions and research projects on Mars. The Ministry of Human Resources and Development will look after the recruitment process and the training of the Brahmavartans. The Ministry of Tourism will take care of future tourists and their needs. The Ministry of Planning will build the infrastructure of MADE and take care of the mission control.

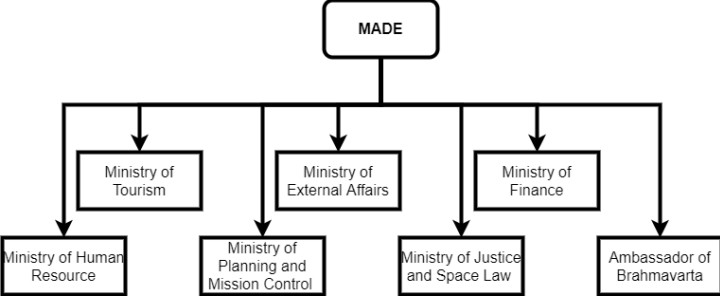

**Figure 1.** Structure of MADE.

*2.2. Settlement Site*

The goal of this design is to build sustainable and enjoyable spaces for Brahmavartans living in extreme conditions, as well as to develop a reasonable community within realistic engineering constraints. Six levels of criteria are used in the settlement site selection.

2.2.1. Selection Criteria

Identification of suitable atmospheric conditions such as temperature, pressure, sunlight, water, and topography. Figure 2a shows climate zones based on temperature, modified by thr topography, albedo, and actual solar radiation, with A = Glacial (permanent ice cap); B = Polar (covered by frost during the winter which sublimates during the summer); C = North (mild) Transitional (Ca) and C South (extreme) Transitional (Cb); D = Tropical; E = Low albedo tropical; F = Subpolar Lowland (Basins); G = Tropical Lowland (Chasmata); H = Subtropical Highland (Mountain). Figure 2b shows the color perspective, with the blue colors indicating high potentials and the darkest blue as the best sites; red colors indicate fewer good sites, with dark red as the worst sites [12].

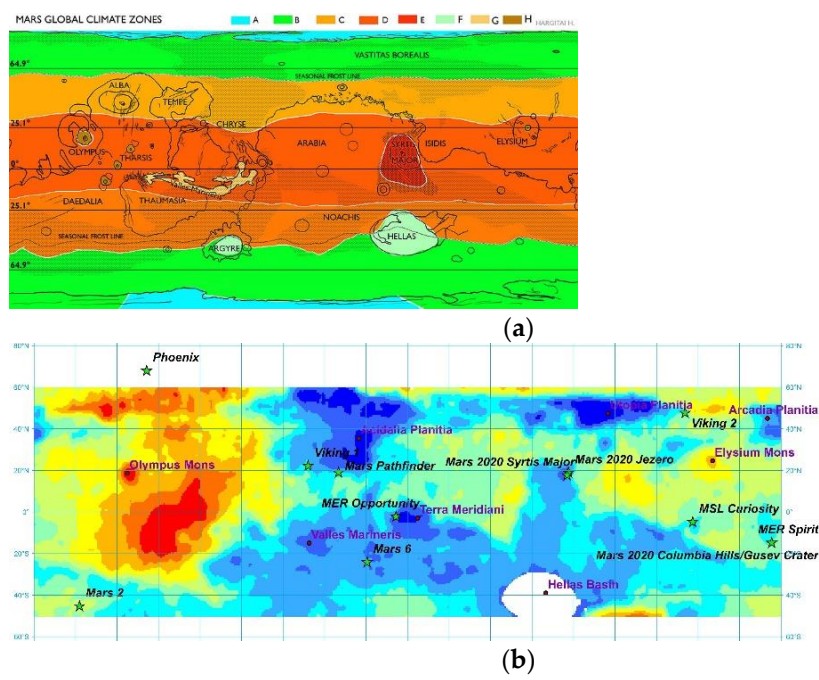

(a)

(b)

**Figure 2.** (**a**) Mars Global Climate Zones, (**b**) Map of the ideal sites from a plant perspective.

2.2.2. Proposed Settlement Sites on Mars

Considering all the six criteria, four sites were chosen to build our states on Mars: Arcadia Planitia, Jezero Crater, Utopia Planitia, and Valles Marineris. Four different states are built with not more than 250,000 Brahmavartans in each state. Two important reasons for building four states are to utilize the resources on Mars to the full extent and for the safety of the Brahmavartans. These four states are named Dvaraka, Hastinapura, Kiskintha, and Indraprastha.

2.2.3. Characteristics of the Selected Site

1.  Arcadia Planitia is flat and smooth, which allows for the construction of the base easier [13]. The subsoil terrain and excess ice are important resources for the base. Its low elevation provides good thermal conditions and solar power. This state is named Hastinapura.

2.  Valles Marineris is an ideal location because of the temperatures (203 K in winters and 313 K in summers), and it is believed that there may be spring-like deposits running beneath the deep canyon, where groundwater could burst through onto the

surface [14]. It has a lower radiation level (15 rem/year) and low altitude. This state is named Kiskintha.

3.  Utopia Planitia is a huge, ~3300 km diameter basin that formed by impact early in Mars's history [15]. This city is named Indraprasta.

4.  Jezero is located close to the Martian equator, providing warmer temperatures than at the poles [16]. There is also evidence of hydrated minerals [17,18]. The availability of water is high, and loss of water from the surface can be minimal due to low elevation as well as a flat ground surface for easy infrastructure development. This city is named Dvaraka.

### 2.3. Dawn of Brahmavarta

In the year 2024, the pre-initiative phase will begin with the selection of the Brahmavartans. In 2026, experiments on soil, communication grids, planning of the missions, and ISRU experiments will begin. The Mars Sample Return mission which will bring Martian soil that will be tested in the labs. This will be followed by sets of cargo missions from 2032 (see Supplementary Table S7 for the details of the Cargo missions to the colony). Initial cargo missions will include the transporting of Nasa Habitat 3D printer [19], Sabatier Reactor [20], and surviving equipment followed by robots, essential raw materials, and equipment for air, and water facilities will be launched by ISRU equipment, spacesuits, and agricultural products. Further cargo missions will include nuclear generators, Rovers, and solar panels followed by medical equipment, raw materials, and manufacturing equipment [21,22]. Initial cargo spacecraft will be transported on a round trip to Mars, by carrying a few tons of hydrogen from Earth and converting it into return trip propellants. By the year 2040, 410 tons of cargo would be sent to Mars to initially start building Hasthinapur.

The training program for Brahmavartans on Earth will start in 2024. Approved space organizations and governments will have to apply, and at full capacity, 3750 trainees will be selected by MADE for each round. The selection will be based on resilience, adaptability, flexibility, the English language, medical history, age between 20–45, and no criminal background. Training will be given on Martian knowledge, software skills, and various group activities. At the end of the cut-off round, the trainee group will be left with 3350 trainees. At full capacity every year, six rounds of training will be considered, and 3350 trainees will go through basic and advanced Martian survival training, as shown in Figure 3. In the end, trainees will be offered to join MADE, for a better understanding of Martian governance. Basic level activities will include technical, physical, and social training. Every candidate will be trained with multiple skills, and points will be awarded to mark their progress. Therefore, the whole training program has been designed for 4 years.

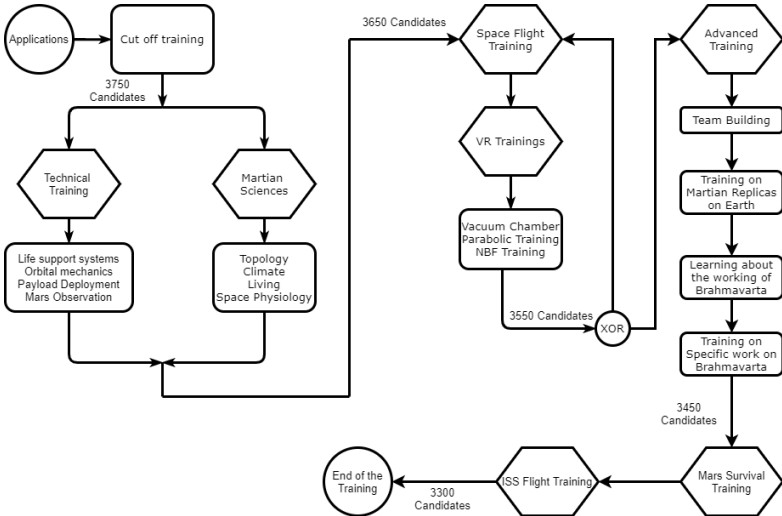

**Figure 3.** Brahmavartans Training.

## 3. Settlement Phase

### 3.1. Brahmavartans' Arrival

In the year 2032, 10 brahmavartans will arrive at Hastinapur. A series of temporary modular pods will be printed from Martian concrete using 3D printing rovers for their stay. In the following years, more Brahmavartans will start to populate the planet. Only engineers from various fields and agronomists will be sent in the first ship, followed by 50 people for the next two every year. Between the years 2034 and 2039, 450 Brahmavartans will populate the planet. From 2040, 10 ships carrying 1346 in total will arrive each year. Considering an improvement in the technology, mortality, and birth rate, it will take approximately 186 years to have a 1,000,000-person colony on Mars. In the early phase, department representatives will be in contact with MADE; once Brahmavarta's population reaches 60%, implementation of the Martian government will be initiated. To avoid an abnormal sex ratio in the future, the male to female ratio will be 50%:49%, respectively. Initially only one state, Hastinapur, will be populated, but later, the population will be equally divided into 250,000 in each state.

### 3.2. Brahmavarta's Architectural Concept

#### 3.2.1. Brahmavarta's Layout

The primary engineering requirements for the design are safety, efficiency, and expandability. Safety requires that there be at least two interconnected and individually pressurized segments. In the case of an accidental loss of pressure, a fire, or another failure, there must be at least one means of backup space. Furthermore, the loss of one space must not cut off the functioning portions of Brahmavarta from each other. Hence, as mentioned in Section 2.2.2, the state of Brahmavarta is divided into four cities, and each city is divided into two sectors. Each sector can accommodate up to 130,000 people including tourists, visiting scientists, and Brahmavartans. Each block is connected physically by a road that interlinks them. All three parameters were kept in mind while developing the Brahmavarta's layout.

#### 3.2.2. Sector Layouts

The total area of a sector is 16 km$^2$. All measurements and calculations to build all the components of this sector were based on [23]. The layout of each sector is designed in an octagonal shape with eight layers, as represented in Figure 4. Each of these layers consists of an essential component required for the genesis and survival of a city and is described in order from the outermost to the innermost. The first layer of the city consists of apartments which consist of two-bedroom flats, as mentioned in Section 3.2.4. These apartments are placed at uniform intervals and are separated from the rest of the layers by a canal, which forms the second layer. The next layer is a necessity for the sustainability of the sector—the food production belt. This layer is interspersed with industrial buildings which are constructed in a manner as to expel the least amount of particulate matter to avoid pollution in the apartment zone. Half of the size of the food production belt is made up of structures to support vertical farming in artificial light [24]. The city state is equipped with a state-of-the-art hyperspectral imaging (HSI) sensor for detecting air pollution [25–27]. The next layer is the energy belt, which contains arrays of backup solar panels affixed to the ground to fulfill the emergency requirements of the sector, along with a few apartment buildings. Beyond this layer of energy generation is the layer which consists of districts with individual housing, which are single bedroom flats. These housings are structurally and aesthetically identical and are designed by the Green Building Construction concepts. The next layer consists of areas of recreation—theaters, pubs, gaming centers, aquariums, gymnasiums, and a multi-purpose sports stadium. They are placed strategically close to the individual housings to encourage congregating masses. Going further inside the sector, the next layer also houses industrial buildings manufacturing other essential requirements of the city. The outer edge of this layer consists of studio-type apartments where people working in the industries would live for emergencies. The studios are built in such a

way that two studios can be combined into one, forming a double-bedroom apartment if necessary. The penultimate layer consists of buildings dedicated to the fine arts and educational institutions. It consists of research labs for various engineering and medical fields. The hospitals are equipped with the state-of-the-art biosensors and AI diagnostic abilities [28–34]. The Brahmavarta Congress is located at the central dome of the sector. The innermost layer also contains a transportation hub, where a public transport vehicle commences its journey before traveling throughout all the layers linearly and radially, plying across all the roads connecting the different layers of the sector. The black lines represent the roads. Amalgamating with another identical sector, the city is constructed.

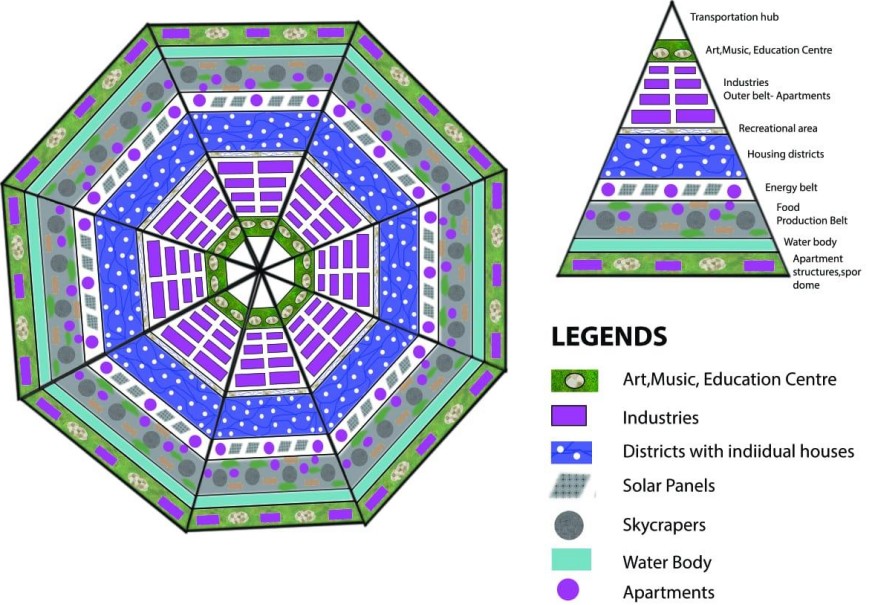

**Figure 4.** Layout of a block.

### 3.2.3. Brahmavarta's Vernacular Architecture

Mars has a thin atmosphere and no global magnetic field, so there is little protection from harmful radiation [35]. The thin atmosphere also means there is little air pressure, so liquids quickly evaporate into gas; despite freezing temperatures, an unprotected human's blood would boil on Mars. However, it would not feel like home if Brahmavartans are walking in space suits in and around the sector. Thus, local terraforming is important for them to feel like they are on Earth. Marine ecology has played an important role in terraforming Earth [36]. To maintain a comfortable temperature and habitable air pressure, each layer of the sector would be made up of pressurized and interconnected bio-domes, each covered with a multilayer transparent polyethylene. The layers in the order from the outermost to innermost are the acrylonitrile butadiene styrene, fluorinated ethylene propylene, crosslinked ethylene tetrafluoroethylene, and polytetrafluoroethylene. These domes are built by tying pressurized inflatables onto the ground, and then the building can be built using the Martian regolith. These layers will protect Brahmavartans from harmful radiations, and inside these domes, the process of Para terraforming can be completed.

### 3.2.4. Living Quarters

The living quarters are divided into three different styles depending upon each Brahmavartan's preference: studio, one-bedroom, and two-bedroom homes, as shown in Figure 5. The size of a studio is 76.03 m$^2$, in which the living room, bedroom, and kitchen are combined into a single room. The studios serve as a home for the bachelors who work in the industries. Since the studios are built next to the industries, people who take care of the emergency needs of the industries will be staying here. The size of a one-bedroom flat is 136 m$^2$. A one-bedroom flat is a little bigger than the studio, which has an extra

washroom and a separate bedroom. These types of flats will be built in the second phase of the layout, as shown in Figure 4. The size of a two-bedroom flat is 163 m². These will be built on the outer zone of the colony in a four-story building.

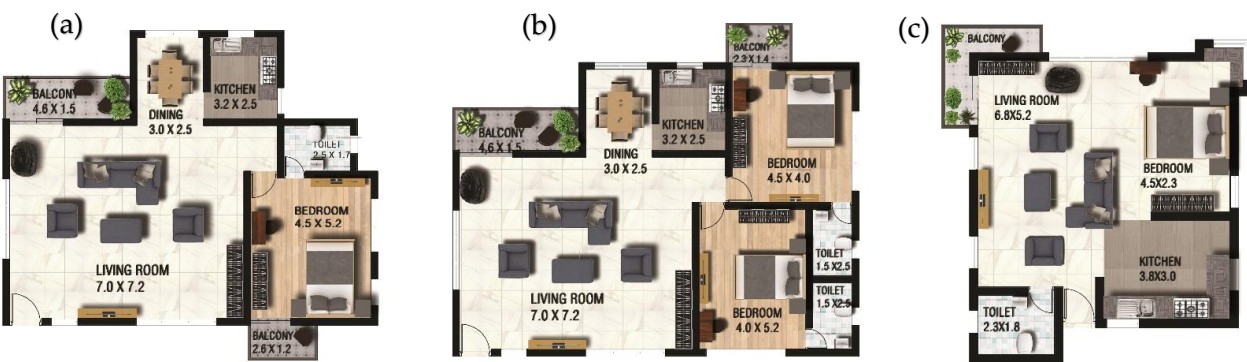

**Figure 5.** (**a**) A studio, (**b**) One Bedroom Apartment, (**c**) Two Bedroom Apartment.

### 3.2.5. Infrastructure Materials

Transporting all the necessities for the infrastructure from Earth is not feasible. The regolith available on Mars is rich in oxides of iron, silicon, magnesium, and aluminum, which can be synthesized, and it is possible to extract essential materials for building infrastructure on Mars. With the abundance of $CO_2$ in Mars's atmosphere, polyethylene, polypropylene, and polycarbonate can be easily manufactured on Mars. A thermomechanical coating (TMC) can be extracted from Mars regolith, along with the bricks and sulfur-based concrete [37]. The thickness of the outside walls will be ~250 mm, of which 235 mm will be brick thickness, and the remaining will be TMC, which can be seen in Figure 6. TMC protects from excessively cold temperatures, impacts, fire, heat, wear, abrasion, chemical degradation, thermal conduction, and radiation heat losses and is also flexible enough to allow easy and repeatable folding and unfolding [5]. For the roof of the greenhouse and inner living quarters, polymethyl methacrylate glass will be used. PMMA is an economical alternative to polycarbonate when tensile strength, flexural strength, transparency, polishability, and UV tolerance are more important than impact strength, chemical resistance, and heat resistance. By extracting and synthesizing the Martian Regolith, the PMMA glass can be manufactured. In addition, PMMA glass provides protection from UV radiation for wavelengths around 300 nm, which is within the range in the case of Mars [38]. Apart from the manufacturing of primary materials, ethylene can also be used for the manufacturing of thermoplastic elastomers, which is the primary material for space suits [39]. Carbon, nickel, manganese, aluminum, steel, and other metals required for manufacturing colony equipment can also be extracted and processed [40]. It is to be noted that the buildings as well as the outer shell of the city will be constructed based on these layers.

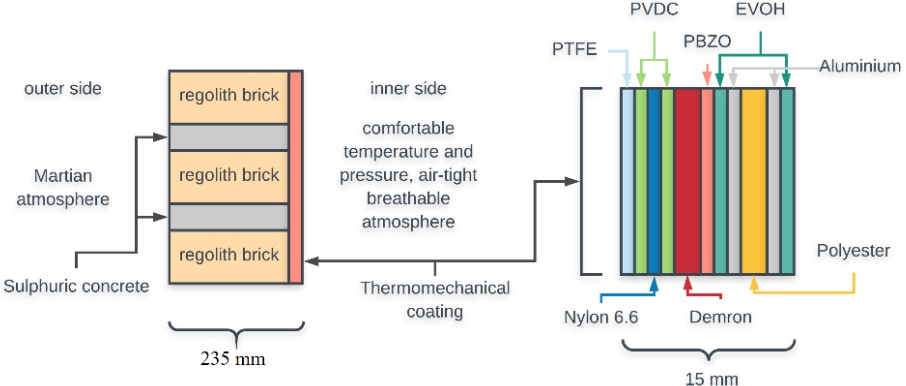

**Figure 6.** Material for outer wall.

*3.3. Transportation*

The four cities of Brahmavarta are located miles apart, and various transportation options for connecting the city's needs are considered and discussed below.

3.3.1. Long and Short Distance Ground Travel

For long distance travel, NASA's Space Exploration Vehicle can be used, and the pressurized cabin can be used for space missions or surface exploration [41]. SEV has the cabin mounted on a chassis, with wheels that can pivot 360 degrees and drive about 10 km per hour in any direction. It can house two astronauts for up to 14 days with sleeping and sanitary facilities, and thus in two weeks, around 3360 kilometers can be traveled [9]. For short distance travel, with not much to carry, electric ATVs can be used [42]. However, the ATVs can also work on the extra methane produced from the Sabatier reactor.

3.3.2. Unmanned Aerial Vehicles

NASA has been testing a flying wing prototype called Preliminary Research Aerodynamic Design to Lower Drag, or Prandtl-D for short, which is capable of flight in the thin Martian atmosphere [43–45]. Ingenuity is a robotic helicopter designed specifically to operate in the low-pressure atmosphere of Mars, which could potentially cover a distance of up to 300 m per flight [46]. The key to its operation, though, seems to be in the shape and size of the counter-rotating rotors. Balloons have been flying for decades in Earth's stratosphere which, at its upper altitudes, has an atmosphere as thin as that on the surface of Mars. Conventional stratospheric balloons, though, only stay aloft for a few days because of the daily heating and cooling of the balloon. Helium super pressure balloons, on the other hand, could fly for more than 100 days, and the same technology can be used for a balloon for Mars [47].

3.3.3. Martian Hopper

Planet hopping is a long-distance travel method that is unlike anything we have on Earth. A 'hopper' takes off like a rocket and begins a ballistic trajectory that shoots it into space, but not into orbit [48–51]. The planet's gravity pulls the vehicle back down to the surface, where it can perform a powered landing, refuel, and do it again. These can be extremely tricky in a heavy gravity environment like Earth, but as both SpaceX and Blue Origin have demonstrated, are clearly doable. On Mars, with only 38% of Earth's gravity, it should be much easier. The Martian Hopper will be powered by a radioisotope thermal generator for the rocket engine, which could pull $CO_2$ from the Martian atmosphere and compress it until it is liquefied. When ready to 'hop', pump the gas into a chamber and expose it to the intense heat from the RTG. A research team from Leicester University and the Astrium space company have proposed calculations suggesting the thrust achieved could enable a one metric ton craft to leap a distance of up to 900 m at a time [52].

*3.4. Brahmavarta's Life Support System*

The schematic of Brahmavarta's life support system (BLSS) consists of a Sabatier reactor from which drinkable water, oxygen, propellant, iron, and steel production and is shown in Figure 7 [53]. The reactor is fed with the atmospheric $CO_2$ and hydrogen electrolyzed from groundwater, which in turn produces oxygen and methane at a ratio of 4:1, and the excess oxygen will be supplied to life support. Considering, the consumption of ~9.6 kg of water per person per day and ~2.8 kg of oxygen per person per day, thus 222 tons of water will be required for 130,000 (including temporary stays) people per day and 364 tons of total oxygen per day. The BLSS water extraction subsystem will be able to extract 68.2 kg/h [6], and hence, 132 facilities will produce 110,635 tons/day. By doing electrolysis, 198 tons/day of $O_2$ is produced, out of which 20,041 kg/day is for propellant generation, and the remaining 3032 kg/day is sent to the air processing plant. From electrolysis, 2922 kg/day of $H_2$ is sent to the Sabatier reactor for producing 5812 kg/day of $CH_4$, which will be used for the propellant generation, and 13,051 kg/day of water

is produced which is sent to the water processing plant. Hence, $9.54 \times 106$ kg/year of propellant is produced, which satisfies the need for incoming and outgoing spaceships. The air processing plant will convert pure $O_2$ into a breathable air ratio. This breathable air will be supplied to living quarters, kitchens, and labs. This air with more $CO_2$ proportion, released from the mentioned areas, will be sent to greenhouse chambers. The plants in greenhouses will produce $O_2$ using the photosynthesis process. This air with more $O_2$ proportion will be an input for the air processing plant (see Supplementary Table S5 for Calculations for Colony Fuel Usage).

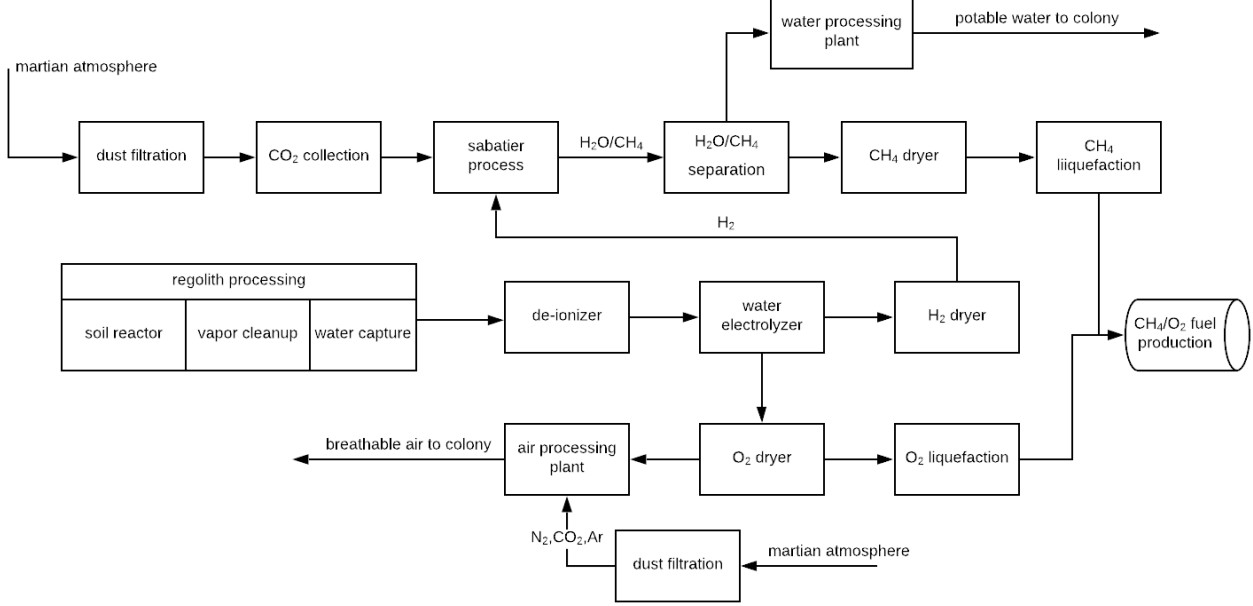

**Figure 7.** Breathable air circulation system.

### 3.5. Electricity Production

For countries that are highly industrialized and have high energy consumption, the amount of energy needed for a city of 1 million is about 1500 MW [54,55]. The consumption of electricity in Brahmavarta will be for domestic and personal uses, scientific and office uses, BLSS, and agriculture. The average electricity consumption for commercial activities is 3.5 kW per person. In addition, power consumption is 17 kW per processing facility [2]. Adding all the needs together with the factor of safety, the power consumption of one sector in a city will be ~600 MWe. Electric power generation will be a combination of uranium-based nuclear reactors, retracting common and solar panels inside the dome, and a system of nickel-hydrogen batteries. The evolution of NASA Kilopower Reactor will result in a 100 kW reactor from the maximum of the currently envisioned design of 10 kW [56–58]. With 200 nuclear reactors, the total power generated is 200 MWe. The roof side of outer living quarters, passage, lab, kitchen, and buildings will be covered with solar panels. The area covered by solar arrays on one sector in buildings will be approximately 75% the size of a sector, i.e., with 33% efficiency, and considering 100 W/m$^2$, the solar power generated will be 400 MWe. Hence, the total power generation will be 600 MWe, satisfying the total electricity need of a sector. Other than these, a series of Mars autonomous and foldable solar arrays will also be deployed outside the sector to charge the nickel-hydrogen batteries to prepare for a worst-case scenario and to power the transportation systems of Brahmavarta.

### 3.6. Food Production and Waste Management

A healthy body requires 2 kg of food every day for nutrition and energy, which includes 1000 mg of calcium, 400 µg of folic acid, and 56 g of protein each day. However, the intake varies, with scenarios such as the iron intake for females being double, while the whole intake doubles in case of pregnancies. Considering these factors, poppy seeds,

winged and soya beans, lentils, potatoes, tomatoes, onions, green beans, spinach, beets, orange, and lemons will be harvested. In addition, every year, a cargo consisting of 100 tons of meat will be imported from Earth to Brahmavarta. These crops require only 7 h of sunlight and 14 h of artificial light, manure, and a proper drainage system. These plants only require a sprinkling of water from time to time. Soil is required to be rich in potassium, phosphorous, and nitrogen nutrients, which can be easily achieved by adding composites, lime, and sulfur to maintain the pH balance for the respective crops. Rhizobium bacteria will be used to make nitrogenous-rich soil [59]. For the first two years, 100 kg of manure will be imported from the Earth, which costs $1000 for a ton, and simultaneously, the Martian soil will be prepared. The food for the first year will be brought by the Brahmavartans in their cargo. The layout of the water processing and the manure processing is represented in Figure 8. Water treatment facilities will purify urine into struvite, a very good fertilizer that is guaranteed to be free from diseases. Human waste will be collected and sent for the processing which will be used as manure for agriculture along with struvite. It has also been found that the holes that worms dig in the soil aerate the mixture and improve the soil's structure, making it easier for water to penetrate the soil and nourish plants. Through their activity in the soil, earthworms offer many benefits: increased nutrient availability, better drainage, and a more stable soil structure, all of which help improve farm productivity. It has been found that earthworms not only can survive but also reproduce in Martian soil, which can be used to aerate and improve the soil structure [60].

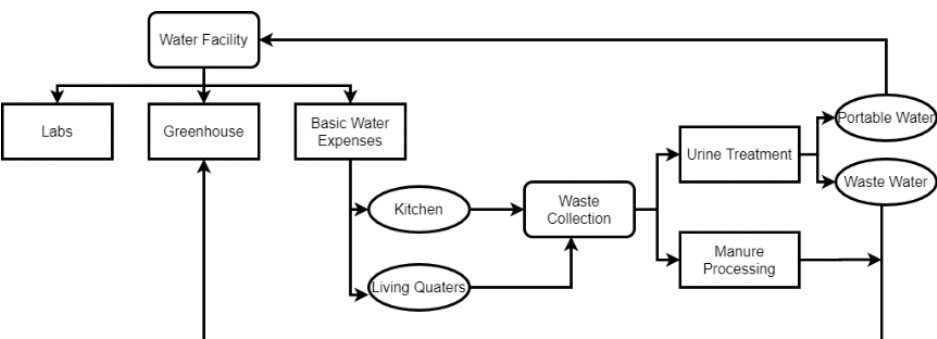

**Figure 8.** Layout of manure and water recycle plant.

### *3.7. Internet*

Internet consists of two big components: first are the servers that store information and their related infrastructure. Providing internet to Mars initially will be difficult and slow. The distance between Earth and Mars at the closest point is 54.6 million kilometers, and the farthest is 400 million kilometers. Thus, for a piece of information to be transferred, it takes around 3–22 min to transfer from Earth to Mars. A laser wave is 100,000 times shorter than a radio wave, which means more room being needed to carry data and a 5 times more reliable connection. A smaller wave means better signal, and fortunately, the technology we need exists which has been tested by the NASA Lunar Laser Communication Demonstration [61–64]. These lasers will be able to handle HD videos and more. NASA is focused on upgrading existing satellites to build a Laser Powered Space Internet Network. With the help of lasers and satellites, the internet works the as same on Earth. However, once the settlement has been completed, an entirely separate set of infrastructure, including servers, will be built on mars, which will be working independently.

### *3.8. Manufacturing Industries*

Iron—The MER-B rover, landing at the Meridiani Planum site, found abundant hematite in the form of small (4–6 mm) spherules [65,66]. These are approximately 50% hematite by weight. Similar to Earth ore deposits, the ore on Mars was concentrated and produced by aqueous diagenesis, at a time early in Mars' history. This is an easily accessible ore; the hematite concretions exist both in the soil, where it can be simply scooped up, and

embedded in soft sedimentary rock, from which it can be easily extracted. The standard terrestrial carbothermic process can be used to reduce hematite to metallic iron since carbon monoxide is easily available at all locations on Mars from the carbon dioxide atmosphere.

Magnesium—In the process of making cement, large amounts of MgO are produced. Magnesium metal is made via the Pidgeon process, whereby MgO is reacted with silicon metal at high temperatures to produce magnesium and $SiO_2$ [67]. The normally unfavorable thermodynamic equilibrium is driven toward completion by distilling away magnesium vapor as it forms.

Aluminum—In the process of making cement, a large amount of $Al_2O_3$ and MgO are produced. Aluminum is produced via the FFC Cambridge process without the need for cryolite ($Na_3AlF_6$), which is difficult to obtain on Mars due to the relative scarcity of fluorine [68]. Instead, calcium chloride ($CaCl_2$) is used to dissolve the alumina, and due to large quantities of chlorine available from perchlorate, this material is readily produced.

Steel—The production of steel, rather than simply iron, requires the addition of carbon in controlled quantities [69,70]. The amount of carbon needed is small, and carbon is easily available from the Martian atmosphere. Carbonyl processing uses carbon monoxide, which is easily manufactured from the Martian atmosphere, to transport the metals in the form of volatile iron- and nickel-carbonyl. Distillation can then separate the iron and nickel, allowing the composition to be modified as desired.

Organic chemicals and plastics—Ethylene can be produced from $CO_2$ and water produced from BLSS using intermittent sunlight. Once Ethylene has been created, a wide variety of more complex molecules can be created. Polyethylene, terephthalate, polyethylene, polypropylene, polyvinyl chloride, polystyrene, polycarbonate, polymethyl methacrylate, polyoxymethylene, acrylonitrile butadiene styrene, and nylon can be synthesized from about three dozen reagents made from simple organic precursors. Together, these polymers enable a wide array of uses and significantly reduce the colony's Earth dependence.

Basalt—Basalt is very common on Mars. It is an igneous rock that is commonly formed by magma extrusions during a lava flow. Basalt is more elastic than steel and has a similar elasticity to aluminum. Fused Deposition Modelling can be performed by using a computer's numerical control, to precisely force molten material through a die to specific places in layers. The molten material flows onto existing material, and thereby, the new and existing material fuse together as the material freezes or solidifies. This approach is commonly used to print thermoplastics such as polylactic acid or acrylonitrile butadiene styrene [71–73].

Concrete and Cement—Cement is produced through the combination of lime (CaO), silica ($SiO_2$), and alumina ($Al_2O_3$) with smaller amounts of magnesia (MgO). All of these compounds are found in Martian regolith, but with much lower amounts of CaO and $Al_2O_3$ than required. By first heating the regolith to separate the lower-melting FeO, and to a lesser extent, $Al_2O_3$, MgO, and $SiO_2$, a mixture with a composition of approximately 52% CaO, 32% $Al_2O_3$, 15% SiO2, and 2% MgO is obtained [74–76]. This cement is then mixed with coarsely crushed regolith and water to produce concrete.

Mirrors and Glass—Using lunar regolith simulant and heating it within a susceptor-assisted microwave oven, it is possible to manufacture a variety of basaltic glasses. Furthermore, it is also possible to shape these glasses by grinding and polishing the surface flat and smooth. Glasses manufactured from different lunar regolith simulants can be coated with aluminum or silver to create reflective properties. With a porous and/or smooth surface finish, mirrors can be made to reflect the incident solar light (400 nm–1250 nm) in between 30% for the worst and 85% for the best samples [77–79].

### 3.9. Communication

Communication between Earth and Mars is conducted via communication channels that orbit Mars. For cases when the Sun eclipses the straight communication, the kind of antennas may be placed in Lagrangian points of the solar system, which will provide robust communication channels with the Earth. Communication within the colony is arranged based on the G5 network with the Ethernet and MQTT protocols that are used

for IoT and smart cities, ensuring all the devices are connected to the web and provide real-time information about their state. IoT also gives a chance to various kinds and uses AI algorithms to optimize the intake of water and food and reduce the other important factors. For better GNSS navigation kinds of communication, posts are printed from the 3D printers to improve navigation for Mars exploration. Brahmavartans will use localized cell networks for the majority of communications. Rovers and EVA suits will be fitted with Bluetooth or wired connectivity. Every colonist will carry a smartphone and wear a smartwatch which will provide instant communication between colonists, tracking capabilities, and health monitoring. Multiple surface towers provide redundancy and geolocation with their independent solar fields. Mars has a sufficient ionosphere to bounce a signal beyond the line of sight, and hence, self-tuning HF radios will be used for backup and emergency communications.

*3.10. Terraforming Mars*

Paraterraforming involves the construction of a habitable enclosure on a planet which eventually grows to encompass most of the planet's usable area. There are three sources of $CO_2$ on Mars [80]. There is the south polar ice cap, which consists of water ice interspersed with a thick layer of $CO_2$ ice [81,82]. The next accessible source is absorbed into the surface dust, the regolith, up to 100 m deep. The final source is carbonate minerals in the crust. To terraform Mars, there are three requirements: warm the planet to 290 k, increase the atmospheric pressure, supply 240 mbar of breathable oxygen, and provide sufficient water for a water cycle. A magnifying Solette and mirror lens can be used to vaporize the Martian regolith by placing a Solette 1000 km from Mars, between the planet and the sun, held in position by light reflected from the annular support mirror. Sunlight from the Solette must be further focused by a lens ranging some 400 km above the Martian surface. Since this is within the upper reaches of the Martian atmosphere, an aerial lens is used instead of orbital mirrors. If the oxygen is liberated through the pyrolysis of regolith oxides or nitrates, no further action will be required. However, any excess carbon dioxide will have to be converted into oxygen by photosynthesis [83]. In addition to this technique, albedo modification, the impact of ice asteroids, and the release of artificial greenhouse agents will be considered and researched after the full colonization of Mars. Once Mars has been terraformed, we need to protect the new atmosphere. We cannot restart Mars's magnetic field, so we can try to build an external magnetic shield. The easiest would be to do that in space by placing an orbiting field generator placed between Mars and the sun. Although, these processes are not as simple as explained, terraforming Mars will not be achieved during the time-frame of the mission. However, by following the procedures mentioned, the process could be fastened.

## 4. Self-Sustaining Phase

To create a self-sustainable environment, a trading system similar to that of Earth has to be introduced which must be profitable in the long run. The important objective of the economic model of Brahmavarta is to check the feasibility of the project, which starts with the generation of the investments. After the initial investments are brought in for the project, the Pre-Initiative phase starts, followed by the settlement phase. The cash inflow–outflow of the project and the consumables in Brahmavarta will be calculated in US dollars since the initial investment is generated on Earth. The important constraint on the case is the transport of goods from Earth to Mars costing $500/kg, and from Mars to Earth, $200/kg. By using these constraints, the economic model of Brahmavarta was developed. For ease of calculation, the inflation has not been considered. The rate of the price of goods has been fixed for the initial few years. However, with the increase in inflation on Earth, the cost of the prices of the goods will grow.

*4.1. Initial Investment*

The foundation of Brahmavarta will be laid through a series of investments from several government bodies on Earth. A total of USD 4 billion from the investors will

be drawn as loans every year. Furthermore, the profits generated for the training of Brahmavartans will also add to the initial account, which is USD 4 million per person. Hence the total initial investment for the project is USD 240 billion.

Generation of Initial Investment—The foundation of Brahmavarta will be laid through a series of investments from across several government bodies on Earth. USD 4 billion from the investors will be drawn as loans every year. Furthermore, the profits generated for the training of Brahmavartans will also add to the initial account, which is USD 4 million per person. Hence the total initial investment for the project is USD 240 billion.

Inflow and Outflow during the Pre-Initiative Phase—The major outflow in the pre-initiative phase will involve 33 cargo missions, with each of them comprising of NASA Habitat 3D printer, Sabatier Reactor, food for the first few Brahmavartans, ISRU equipment, spacesuits, agriculture materials including seeds and fertilizers, nuclear generators, rovers, solar panels and medical equipment, and various other items required for a settlement.

Inflow and Outflow during the Settlement Phase—The settlement phase will comprise a total of 156 missions. A total of 20,176 tons of cargo would be sent at the end of the settlement phase. The first Brahmavartans will be sent with cargo having food and various utilities for their survival. These values are rough estimations concerning the weight distribution. During this phase, a few of the profitable operations mentioned in Section 4.2 will begin, thereby bringing in an inflow of cash to the economic model of Brahmavarta.

### 4.2. Profitable Operations

The economic model of Brahmavarta is based on the model of the trading system as on Earth, and to keep the economy self-sustaining, this economy model discusses various profitable operations and estimates the profit generated upon their installation in a fully functioning settlement. The proposed methods of sustenance on Mars are as follows:

#### 4.2.1. Asteroid Mining

An abundant source of rare metals of rich quality can be found in the Asteroid belt near Mars [84–86]. Asteroids such as Aspacia, Rundra, etc., help us estimate revenues that could be generated from asteroid mining. Since the mass ratios in the calculation of fuel expended in asteroid mining are not determined for these individual asteroids yet, we took the mass ratio from Ceres, which is further away from the asteroids in the study. The total payload from the asteroid in one mission is estimated to be 7500 tons, which can be extracted with an efficiency of 75%. The frequency of the missions every year is four, and once every four years, an extra mission will be sent to Earth. The payload in every trip on the rocket is 150 tons, which brings the total amount transported per year to 637.5 tons. Revenue generated through asteroid mining can be seen in Table 1. Assuming the composition by weight of platinum from the asteroid mined to be 25%, the remaining ore is rich in elements such as nickel, iron, copper, etc., which can be utilized on Brahmavarta for different purposes (see Supplementary Table S8 for the details of the Asteroid Mining missions to the colony).

**Table 1.** Revenue from Asteroid Mining.

| Asteroid Mining | | |
|---|---|---|
| Number of Transits | 4 | Per year |
| Platinum Extracted | 637.5 | Tons |
| Selling Price on Earth | 30,000 | Per Kg |
| Cost of Transportation | 752 Million | Per year |
| Estimated Cost of Payload | 19 Billion | Per year |
| Total Revenue Generated | 18 Billon | Per year |

#### 4.2.2. Research Visits

Every year, 85 selected scientists from countries/organizations who have sponsored the initiative would be sent to Brahmavarta for a year of extensive research. They would

be hosted in the city nearest to their area of research. The revenue generated by these research visits would be 891.431 million every year, which is represented in Table 2 (see Supplementary Table S10 for the details of the research visit mission to the colony).

**Table 2.** Revenue from Research Visits.

| | Research Visit | |
|---|---|---|
| Number of Tourists | 85 | Per year |
| Selling Price | 20 Million | Per researcher |
| Profit Generated | 10 Million | Per researcher |
| Total Revenue | 8.91 Million | Per year |

### 4.2.3. Tourist Visit

A total of six schemes have been proposed, A through F are named in order of the number of days of travel, as shown in Table 3. The price of the tickets is directly proportional to the number of days of stay, starting at USD 15 million for 30 days of stay, up to a maximum of USD 30 million for a stay up to 550 days. The aim of having multiple schemes is to increase the frequency of travel to Mars. The available windows of travel for a sample year were found in [87,88]. The gradient in the color of the data demarcates the journeys under different schemes. A total of 1100 tourists can be accommodated a year at all four different states, each hosting different types of schemes on a shifting basis. Revenue of USD 31 billion is generated every year by tourism. One of the assumption considered here is that tourism will be very active in the future (see Supplementary Table S9 for the details of the tourism missions to the colony).

**Table 3.** Revenue from Tourism.

| Plan | Stay Time | Duration | Frequency | Cost/Person | Ticket/Person | Total Revenue/Year |
|---|---|---|---|---|---|---|
| A | 30<br>30 | 465<br>410 | 3 | $6.3 Million | $15 Million | $3.2 Billion |
| B | 112<br>112 | 800<br>688 | 3 | $9.5 Million | $18 Million | $2.1 Billion |
| C | 128<br>288 | 800<br>848 | 3 | $6.5 Million | $21 Million | $5.5 Billion |
| D | 320<br>368 | 896<br>816 | 3 | $10 Million | $24 Million | $3.3 Billion |
| E | 384<br>448 | 912<br>880 | 3 | $7 Million | $27 Million | $7 Billion |
| F | 464<br>544 | 864<br>896 | 3 | $6 Million | $30 Million | $9.8 Billion |
| | | Total Revenue Generated = | | | | $31 Billion |

### 4.2.4. Deuterium Generation

Deuterium, the heavy isotope of hydrogen, is 166 ppm by composition on Earth and comprises 833 ppm on Mars [89–91]. Deuterium is the key fuel for both the first- and second-generation fusion reactors on Earth. On Earth, 1 kg of deuterium is priced between USD 10,000–16,000, depending on the purity. Therefore, the estimated profit per kg is around USD 9500, even if the cost drops due to less demand. For one ton, the estimated profit is USD 9,500,000. As of now, on Earth, 452 nuclear power plants are either used or under construction, which requires 400 g of Deuterium per day for 1500 MW production. Upon calculation, it can be inferred from Table 4, that 43.8 tons of deuterium are required per year. On transporting 90 tons of deuterium once in two years, the profit is USD 746.58 million (see Supplementary Table S3 for deuterium mining cost and plan).

**Table 4.** Revenue from Deuterium exports.

| Total D Exported from Mars | 87,989.33 kg |
|---|---|
| Transportation costs | USD 133.32 million |
| Cost of D on Earth | USD 10,000 |
| The total cost of D on Earth | USD 879.89 million |
| Total Revenue generated in 2 years | USD746.58 million |

4.2.5. Lunar Dust

In 2003, the federal government put a price tag on moon rocks. NASA assessed the value of the rocks at around USD 50,800 per gram in 1973 dollars, based on the total cost of retrieving the samples. That works to over USD 300,000 a gram in today's value. Around 15 kgs of non-transferable Martian dust will be sold to specific research organizations and universities as per their specific needs for research purposes on a rolling basis, as shown in Table 5 (see Supplementary Table S2 for revenue generated from Mars soil).

**Table 5.** Revenue from Lunar Dust.

| Lunar Dust | |
|---|---|
| Cost of Lunar Dust | USD 200 Million/kg |
| Total kgs transported | 15.00/year |
| Total Cost of Transportation | USD 3000.00/year |
| Total Revenue Earned | USD 3 Billion/year |

4.2.6. Broadcasting

In 2018, the total revenue from television broadcasting rights of Fédération Internationale de Football Association (FIFA) was USD 2543.97 million [92,93]. These are the trends in broadcasting on Earth for events of the magnanimity of the moon landings and the Olympics event. Once the settlement is completed, every Friday, Saturday, and Sunday evening, a sporting event will be covered extensively and will be broadcast on Earth. In the pre-initiative phase, selective events of six such as the training, arrival, and departure of Brahmavartans and tourism would be covered every year. As shown in Table 6, while the full functioning of a Brahmavarta, ~USD 30 billion can be earned through sponsorship, ticketing, and TV rights (see Supplementary Table S1 for revenue generated from broadcasting). Every month, an interesting event such as soccer, cricket, and even world chess championship and a Mars version of Olympics could be designed to keep the interest of the broadcasting.

**Table 6.** Revenue from Broadcasting.

| Broadcasting | |
|---|---|
| Revenue Generated | USD 183 Million/event |
| Number of Launch Events On Earth | 6/year |
| Number of Sporting Events on Mars | 156/year |
| Total Revenue Generated in the Pre-Initiative Phase Phase | USD 1 Billion/year |
| Total Revenue Generated in the Self-Sustaining Phase | USD 29 Billion/year |

*4.3. Future Scope and Economic Viability*

In the year 2086, all the initial investments can be paid off, which shows that the economic model of Brahmavarta is viable, since it promises sustenance in 62 years. After 2086, MADE will become the Martian Embassy on Earth (MEE). Brahmavarta can also act as a pit stop for refueling in the future for missions such as Human Outer Planet Exploration. The Martian economy shall also flourish by aiding in bio-medical research,

which is primarily due to the variety in atmospheric conditions that make it possible to grow and treat certain microbes that cannot be on Earth. The semi-conductive material research which requires highly vacuum-like conditions not available on Earth can be smoothly regulated on Mars. There are many trips suggested between Earth and Mars in the economy plan above, which paves the way for selling water from Mars to in-orbit manned facilities. In the future, there will be a triangle trade, with Earth supplying high-technology manufactured goods to Mars, Mars supplying low-technology manufactured goods and food staples to the Moon, and Moon sending helium-3 to Earth [94–96].

Cost Plan

Table 7 shows the summary of the differentiation of the ventures across a span of 60 years, which is studied and analyzed to forecast the self-sustenance point of Brahmavarta (see Supplementary Table S6 for detailed cost budget of the initiative). The cost flow is laid out based on the timeline of the project considering mandatory ventures and those introduced to incur profits. Since all three phases have some profitable operations, it is easier to attain the self-sustaining phase. The break-even analysis of the economic model of Brahmavarta is represented in Table 8. It is clear that after the full functioning of all the profitable operations in 2084, the initial investments can be paid off, and the future profits can be solely used for the development of Brahmavarta (see Supplementary Table S4 for breakeven analysis economic plan of the initiative).

**Table 7.** Summary of Cost Plan for Brahmavarta.

| Cost Budget | | | | | | |
|---|---|---|---|---|---|---|
| Years | Details | Outflow Cost Estimation | Total Outflow | Outflow Margin with Margin | Inflow Estimation | Total Inflow |
| 2024–2034 | MADE member's investments<br>MADE operating cost<br>MADE Infrastructure<br>The training program at MADE<br>Spacecraft Manufacturing | <br>$19 Billion<br>$3 Billion<br>$2.8 Billion<br>$9 Billion | $34.7 Billion | $36.4 Billion | $48 Billion<br><br><br>$7.5 Billion | $55.5 Billion |
| 2036–2046 | MADE member's investments<br>MADE operating cost<br>Spacecraft Manufacturing<br>Broadcasting<br>Manned Mission<br>Cargo Missions<br>Martian Soil<br>The training program at MADE | <br>$19 Billion<br>$9 Billion<br><br>$54 Billion<br>$31 Billion<br><br>$97 Billion | $210 Billion | $221 Billion | $48 Billion<br><br><br>$6.5 Billion<br><br><br>$35 Billion<br>$258.8 Billion | $349 Billion |
| 2048–2058 | MADE member's investments<br>MADE operating cost<br>Spacecraft Manufacturing<br>Manned Mission<br>Broadcasting<br>Cargo Missions<br>Deuterium<br>Martian Soil<br>The training program at MADE | <br>$19 Billion<br>$9 Billion<br>$77 Billion<br><br>$26.3 Billion<br><br><br>$98.8 Billion | $230.5 Billion | $242 Billion | $48 Billion<br><br><br><br>$6.5 Billion<br><br>$4.4 Billion<br>$35.9 Billion<br>$263.6 Billion | $358.7 Billion |
| 2060–2070 | MADE member's investments<br>MADE operating cost<br>Spacecraft Manufacturing<br>Manned Mission<br>Broadcasting<br>Cargo Missions<br>Deuterium<br>Martian Soil<br>The training program at MADE | <br>$19 Billion<br>$9 Billion<br>$77 Billion<br><br>$26.3 Billion<br><br><br>$103.9 Billion | $235.5 Billion | $247.3 Billion | $48 Billion<br><br><br><br>$6.5 Billion<br><br>$4.4 Billion<br>$35.9 Billion<br>$277.1 Billion | $372.1 Billion |

**Table 7.** *Cont.*

| | | Cost Budget | | | | |
|---|---|---|---|---|---|---|
| Years | Details | Outflow Cost Estimation | Total Outflow | Outflow Margin with Margin | Inflow Estimation | Total Inflow |
| 2072–2082 | MADE member's investments | | | | $48 Billion | |
| | MADE operating cost | $19 Billion | | | | |
| | Spacecraft Manufacturing | $9 Billion | | | | |
| | Manned Mission | $106 Billion | | | | |
| | Broadcasting | | $311.3 Billion | $326.9 Billion | $6.5 Billion | $497.3 Billion |
| | Cargo Missions | $26.3 Billion | | | | |
| | Deuterium | | | | $4.4 Billion | |
| | Martian Soil | | | | $35.9 Billion | |
| | The training program at MADE | $150.8 Billion | | | $402.2 Billion | |
| 2084 | MADE member's investments | | | | $8 Billion | |
| | MADE operating cost | $3 Billion | | | | |
| | Spacecraft Manufacturing | $1.5 Billion | | | | |
| | Manned Mission | $19.2 Billion | | | | |
| | Broadcasting | | | | $29.6 Billion | |
| | Deuterium | | $49.1 Billion | $51.5 Billion | $746 Billion | $212.3 Billion |
| | Martian Soil | | | | $5.9 Billion | |
| | The training program at MADE | $25.1 Billion | | | $67 Billion | |
| | Asteroid Mining | | | | $36.7 Billion | |
| | Research Visits | | | | $1.7 Billion | |
| | Tourism | | | | $62.3 Billion | |
| | Total Money Borrowed = | $240 Billion | | Money Left after Payback | $479.9 Billion | |

**Table 8.** A sample of profits generated in the self-sustaining phase.

| | | A Sample Balance Sheet for Every 10 Years | | | | |
|---|---|---|---|---|---|---|
| | Broadcasting | | | | $29.6 Billion | |
| | MADE operating cost | $19 Billion | | | | |
| | Spacecraft Manufacturing | $9 Billion | | | | |
| | Manned Mission | $106 Billion | | | | |
| | Cargo Missions | $26.3 Billion | | | | |
| 2200–2210 | Deuterium | | $286.2 Billion | $300 Billion | $3.7 Billion | $903 Billion |
| | Martian Soil | | | | $29.9 Billion | |
| | Tourism | | | | $311.8 Billion | |
| | Research Visits | | | | $8.9 Billion | |
| | Asteroid Mining | | | | $183.7 Billion | |
| | The training program at MADE | $125.7 Billion | | | $335.2 Billion | |
| | | Total Outflow = | | $300.5 Billion | Total Inflow = | $903 Billion |

## 5. Social and Political Model

### 5.1. The Brahmavarta Congress (TBC)

As the population of Brahmavarta increases, a government will be formed that consists of ten ministries. Exactly when the population of a state reaches 50%, the members of this Congress will be chosen by the people using a democratic system. The Congress is a flat hierarchy model similar to the system of boards. Ten different board members will be elected to head the ministries. These ten ministries, along with the six ministries in MADE, will vote on the new policies to implement on Brahmavarta. The ten ministries include the Ministry of Education, Ministry of Health Care, Ministry of Science, Ministry of Food and Agriculture, Ministry of Labor and Employment, Ministry of Internal and External Affairs, Ministry of Culture and Sports, Ministry of Happiness, Ministry of Finance, and Ministry of Industries. Each representative will maintain discipline and check the functionality of Brahmavarta. The ten ministries and their members will constitute the Brahmavarta Congress The responsibilities of the Ministry of Education are to draw up and improvise strategies, policies, and plans for educational reform and development at various levels of the school; and to draft relevant rules and regulations and supervise their implementation. The responsibilities of the Ministry of Health Care include a contribution to socio-economic development and the development of a local health industry by promoting health and vital-

ity through access to quality health for all Brahmavartans using motivated personnel. The responsibilities of the Ministry for Agriculture and Food cover land utilization, agriculture and forestry, and the development of new agriculture-based industries. Its objective is to safeguard the agriculture resource base, extend the industry knowledge base, and enable nationwide economic growth and employment based on agriculture and agriculture-based products. The main responsibility of the Ministry of Labor and Employment is to protect and safeguard the interests of workers with due regard to creating a healthy work environment for higher production and productivity and to develop and coordinate vocational skill training and employment services. The Ministry of Internal and External Affairs is responsible for the city's security, emergency management, supervision of other ministries, the conduct of elections, public administration, and immigration matters. The Ministry of Culture and Sports is responsible for developing and sustaining ways and means through which the creative and aesthetic sensibilities of all people remain active and dynamic. It also takes care of the promotion of sports and games and is responsible for conducting various tournaments of all the popular sports. The Ministry of Happiness works to improve the levels of happiness in the country through a variety of policies measuring the effectiveness of a congress's various social welfare programs. The Ministry of Finance manages all the government's financial assets and smart cards, proposes economic and financial policy, and coordinates and supervises all profitable operations. The Ministry of Science works to promote new areas of science and technology and plays the role of a nodal department for organizing, coordinating, and promoting science and technology activities in the country. The Ministry of Industry overlooks and regulates the working of trade and industry and is responsible for the growth and development of the industry. TBC also has an Earth Ambassador, who overlooks the council but will not participate in the decision-making process. TBC will communicate with the other six ministries on Earth. Once Brahmavarta attains self-sustainability, TBC will become independent from Earth's finances. With increase in the required mobility in the solar system through direct fusion drive [97], the connect between Mars and Earth making the transportation easier.

*5.2. Smart Card System*

The flow of money within Brahmavarta will be in the form of credits. Brahmavartans will be given a handy card with their bio-metric data imprinted in a chip. This card will be recharged every day with credit points based on their working hours. Brahmavartans will be working 8-h-long shifts, at the end of which they will be given 240 credit points. Independent of the job, each worker will get 30 credits per hour. These credit points, which are stored in a smart card, will be used for buying basic utilities and other entertainment. With an average living standard, a Brahmavartan will require 126 points each day. So, at the end of the day, approximately 114 points will be saved. The saved points can be used during days of casualty or in sickness.

## 6. Conclusions

In this study, a design plan was created for a working Mars city state of 1 million people, which would be self-supporting to the maximum extent possible including the ability to produce essential bulk materials such as food, fabrics, steel, glass and plastics on Mars, and fabricate them into useful structures, so that 3-D printing and other advanced fabrication technologies and relying on a minimum mass of imports from Earth were included in the design. The city state developed in this study will be able to produce all the food, clothing, shelter, power, common consumer products, vehicles, and machines for a population of 1,000,000, with only the minimum number of key components, such as advanced electronics, needing to be imported from Earth. The major factors considered during this study are technical, economic, socio-cultural and political design.

**Supplementary Materials:** The following supporting information can be downloaded at: https://www.mdpi.com/article/10.3390/universe8110550/s1, Table S1: Revenue generated from broadcasting; Table S2: Revenue generated from Mars Soil; Table S3: Deuterium mining cost and plan; Table S4: Break even analysis economic plan of the initiative; Table S5: Calculations for Colony Fuel Usage. Table S6. Detailed cost budget of the initiative; Table S7: The details of the Cargo missions to the colony; Table S8: The details of the Asteroid Mining missions to the colony; Table S9: The details of the tourism missions to the colony; Table S10: The details of the research visit mission to the colony.

**Author Contributions:** Conceptualization, A.M.; data curation, A.M.; formal analysis, A.M.; funding acquisition, A.M. and H.-C.W.; investigation, A.M.; methodology, A.M.; project administration, A.M. and H.-C.W.; resources, A.M.; software, A.M.; supervision, A.M. and H.-C.W.; validation, A.M.; writing—original draft, A.M.; writing—review and editing, A.M. and H.-C.W. All authors have read and agreed to the published version of the manuscript.

**Funding:** This research was supported by the National Science and Technology Council, The Republic of China, under the grants NSTC 110-2634-F-194-006, and 111-2221-E-194-007. This work was financially/partially supported by the Advanced Institute of Manufacturing with High-tech Innovations (AIM-HI) and the Center for Innovative Research on Aging Society (CIRAS) from The Featured Areas Research Center Program within the framework of the Higher Education Sprout Project by the Ministry of Education (MOE) in Taiwan.

**Institutional Review Board Statement:** Not applicable.

**Informed Consent Statement:** Not applicable.

**Data Availability Statement:** Not applicable.

**Acknowledgments:** This study is the response to the Mars Society's international contest for the best design plan for a Mars city-state of 1,000,000 people. It was also one of the 20 semi-finalists. The list of semi-finalists can be found at https://www.marssociety.org/news/2020/08/31/semi-finalists-chosen-for-mars-city-state-design-contest/ (accessed on 22 October 2022). It has also been published in the book Mars City States: New Societies for a New World.

**Conflicts of Interest:** The authors declare no conflict of interest.

## Notes

1　The term refers to the area as the place where the "good" people are born. The name can be translated to "holy land", "sacred land", "abode of gods", and "the scene of creation"

2　The Citizens of Brahmavarta

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
