# Peer review of "The Brahmavarta Initiative: A Roadmap for the First Self-Sustaining City-State on Mars"

_universe, doi:10.3390/universe8110550_

Round 1
Reviewer 1 Report
The proposal to build colonies on Mars is absurd. However, this paper was in response to a solicitation to consider such colonies, so it should be published, even if only to show how absurd they are.
Minor comments:
A large rocket like Space X's Falcon Heavy might be capable to landing a ton on Mars (the most massive Martian missions to date have landed a few hundred kg), at a cost of about $1 billion, or about $1 million/kg. That's a rough estimate, but probably right to within a factor of two. The estimate on l. 66 of $500/kg is low by at least a factor of 1000. It is fantasy.
Similarly, the speculation on l. 59 that it will be possible to deliver 100 tons to Mars is fantasy.
The radiation level (l. 128) of 15 rem/year is three times the dose rate allowable for nuclear industry workers, 50--100 times typical background rates on Earth, and an even greater multiple of the allowable dose to the general population.
l. 60: Liquid oxygen and liquid methane require continual expenditure of energy for refrigeration. They are fine for the initial launch and orbital insertion, but cannot feasibly be carried to Mars.
l. 538: assumes an asteroid to be 25% platinum. Platinum is a trace constituent (perhaps a millionth of a percent) of metallic asteroids. Fantasy is no substitute for knowledge.
l. 563: Deuterium is available in unlimited quantity on Earth. There is no need to provide an extraterrestrial source, and these authors to not address the question of how to extract it from Martian material where is is present in only trace quantities (even hydrogen is only present in trace quantities on Mars because it is a dry planet that has lost almost all its water).
Reviewer 2 Report
Review of Paper for MDPI Universe:
The Brahmavarta Initiative: A Roadmap for the First Self-Sustaining City-state on Mars
By Arvind Mukundan, and Hsiang-Chen Wang
General comments:
· This paper about states and cities on Mars is very futuristic and contains a lot of visions and speculations. Nevertheless, it may be useful for defining long term research and funding routes.
· I’m missing several basic engineering and scientific caveats concerning large scale space transport (with its environmental impact, e.g. first stage rocket propulsion pollution).
· More details on the inflation scheme used for long term predictions should be given.
· Budgets: I think one should use scientific notation (or million, billion etc.) for the long numbers in the cost tables. Readability and text space usage will be much improved.
· 3.10 Terraforming Mars: I have serious doubts that such processes are so simple, and, if feasible, they could not be done within the timeframe of the Brahmavarta project! More critical comments must be added.
· A comprehensive conclusion is missing.
Detailed comments, text comments, typos, etc.
Line 45 ... mountains, canyons and deserts …
Line 66 ... 500/kg ... units needed
Figure 2 increase readability (text size)
Line 139 …. or soil …
Line 144 … and agricultural products
Line 146 … raw material ..
Line 170 technology, mortality, and birth rate,
Figure 4 it is not clear to me, if buildings have individual outer shell protections or if a shell hosts many apartments etc.
Line 235 … para terraforming… explain this term a bit more!
Figure 5 mark a) b) and c) in the sketches
There is a second figure 5 correct numbering, and change parenthesis with 235mm
Line 264 … Martian regolith …
Line 309 … propellant, iron
Line 337 ... will we really use nickel hydrogen batteries in that timeframe?
Line 337 … NASA Kilopower reactor
Line 344 … series of Mars autonomous and foldable solar arrays will…
Line 363 … 100 tons of meat ...
Line 371 … water processing and manure processing..
Line 374 … agriculture ..
Line 379 … that earthworms ..
Line 386 … transfer from Earth to Mars. A laser wave
Line 390 … These lasers will be able to handle HD videos and more
Line 391 …lasers ...
Line 422 … using intermittent sunlight ..
Line 422 … Ref [67] is not about using sunlight!
Line 454 ... which provide robust communication channels …
Line 456 will we really use G5 network and MQTT protocol in a far future?
Line 470 … there are three sources (numbers should be written as text if < 10)
Line 473 … there are three requirements; -warm the planet to 290K, increase…
Line 497 add $/kg
Line 504 … 240 billion US$
Line 511 … will involve 33 cargo mission …
Line 512 … Sabatier reactor
Line 514 … generators, rovers …
Line 529 … Rundra, etc. help us …
Line 543ff It is not clear what kind of revenue will be generated by scientists traveling to Mars. Trips may be payed by research organizations, but usually on a cost basis. Please explain.
Line 548 Mars tourism: please explain the assumption, that tourism will be very active in future. The interest in a Mars colony may vanish. The price is high, as are the risks, and the attractivity of being “one of the few” may be gone.
Line 567 Please explain the meaning of the statement of nuclear reactors “as for now”, which use deuterium.
Table 4 check units: specific costs of D …
Line 575ff Lunar dust: it is not clear how the arguments concerning Lunar dust apply to Marsian soil, and what is the benefit in transporting it to Earth?
Line 584ff Broadcasting. I have doubts that in the end there will be a high permanent interest in broadcasting news from Mars, unlike special events like soccer or the first man on the Moon. Who is watching channels from a permanent Antarctica station? The numbers in the paper should be checked!
Line 590 … selective events of six such as training, arrival, departure …
Line 593 … and TV rights.
Tables 7 and 8 it’s very hard to check numbers (see other comment on numbers)
Line 616 … conditions that makes it possible to … that cannot be done on Earth.
Line 636 … Brahmavarta Congress.
Line 640 … Ministry of Health Care …
Line 679ff … 5.3 Worst case scenario: this text seems very simplistic and naïve concerning crimes and riots. Furthermore, if Brahmavarta is independent of Earth, what about “just send people back to Earth? Either leave that part out or elaborate it more.
Line 706 … was financially/partially ...
Line 707 … High-tech Innovations …
References check formatting of references: at least 10 occurrences of years not being bold etc.
